

# Two species of Thoracostomopsidae (Nematoda: Enoplida) from Jeju Island, South Korea

Raehyuk Jeong[1], Alexei V. Tchesunov[2] and Wonchoel Lee[1]

[1] Department of Life Science, Hanyang University, Seoul, South Korea
[2] Department of Invertebrate Zoology, Moscow State University, Moscow, Russia

## ABSTRACT

During a survey of intertidal zones at beaches on Jeju Island, two species belonging to the family Thoracostomopsidae were discovered. One new species, *Enoploides koreanus* sp. nov. and one known species, *Epacanthion hirsutum Shi & Xu, 2016* are reported. Along with morphological analysis, mitochondrial cytochrome oxidase c subunit 1 (mtCOI) sequences and 18S rRNA sequences of the species were also obtained and used to check relative p-distance and phylogenetic positions. While most species of *Enoploides* have long spicules, the new species belongs to a group of *Enoploides* with short spicules < 150 μm). Of the seven species with short spicules, the new species is most closely related to *E. disparilis Sergeeva, 1974*. They both have similar body length, fairly similar sized and shaped spicules with small gubernaculum running parallel to distal end of spicule, and an index value of b. The new species can be distinguished from *E. disparilis* by having pre-anal supplementary organ with short conical tail, while *E. disparilis* lacks pre-anal supplementary organ and has a long conico-cylindrical tail. Along with the description of the new species, the genus *Enoploides Ssaweljev, 1912* is bibliographically reviewed and revised. Of 45 species described to date, 27 are now considered valid, 16 species inquirendae due to inadequate descriptions and ambiguity of the material examined, along with two cases of nomen nudum. With this review, we provide an updated diagnosis and list of valid species, a tabular key comparing diagnostic characters of all valid species, and a new complete key to species. One known species, *Epacanthion hirsutum Shi & Xu, 2016*, is reported in Korea for the first time. The morphology agrees well with the original description provided by *Shi & Xu, 2016*. As they had already reviewed the genus at the time of reporting four *Epacanthion* species, we provide only a description, depiction, and measurements for comparison purposes.

# INTRODUCTION

The family Thoracostomopsidae *Filipjev, 1927* consists of three subfamilies: Thoracostomopsinae *Filipjev, 1927* (two genera), Trileptiinae *Gerlach & Riemann, 1974* (one genus), and Enoplolaiminae *De Coninck, 1965* (19 genera). They are distinguished by the presence or absence of mandibles (Enoplolaiminae or Trileptiinae respectively) or by the presence

Corresponding authors
Alexei V. Tchesunov,
avtchesunov@yandex.ru
Wonchoel Lee, wlee@hanyang.ac.kr

of a long and eversible spear (Thoracostomopsinae). *Enoploides Ssaweljev, 1912*, belonging to Enoplolaiminae, was first erected with type species *Enoploides typicus Ssaweljev, 1912* from Russia. The genus is characterized by its high lips with striation; Y-shaped mandibles consisting of two lateral bars converging into one solid bar with a claw-like distal end, curving inwards to the lumen; onchia that are usually shorter than the mandibles; and spicules that are typically long armed with gubernaculum. Multiple revisions and updates of the genus have taken place. The most notable revision was made by *Wieser & Hopper (1967)*, who questioned validity of many existing species. They argued that classification in this genus is only possible according to male genital armature (gubernaculum), and that all description based only on females and juveniles be considered species inquirenda. Aside from this, species have continuously been transferred from and to closely related genera such as *Enoplus Dujardin, 1845*; *Enoplolaimus De Man, 1893* and *Epacanthion Wieser, 1953*. Most recently, *Smol, Muthumbi & Sharma (2014)* listed 28 valid species including three transferred from *Epacanthion* by *Greenslade & Nicholas (1991)*. By the number, it is clear that *Smol, Muthumbi & Sharma (2014)* considered and applied the list of species inquirendae provided by *Wieser & Hopper (1967)*, but a full list of species was not supplied. A species list provided by NeMys (*Bezerra et al., 2019*) still lists some of the species considered invalid by *Wieser & Hopper (1967)* as valid species; contains erroneous species such as "*Enoploides uniformis* (*Pavljuk, 1984*)" (discussed further in results); and is missing two of three species transferred from *Epacanthion* by *Greenslade & Nicholas (1991)*. Our most recent list of valid species consists of 27 species, including the new species being reported. The last report on *Enoploides* species dates back to 1993, with a new report on *Enoploides stewarti* (*Nicholas, 1993*) from a freshwater lake in South Australia. Most species of the genus are from marine habitats with the exception of two freshwater species (*Enoploides fluviatilis Mikoletzky, 1923* and *E. stewarti*). Of the 27 valid species, 63% (17) were initially reported from Europe; 14.8% (4) from North America; 7.4% (2) from Asia (including the new species); with 3.7% (1) each from South America, Africa, Australia and the Arctic.

The aim of this study was to review and revise the genus *Enoploides* while reporting a new species, *Enoploides koreanus* sp. nov., found from Jeju Island, Korea. *Epacanthion hirsutum* originally reported from East China Sea, is also reported in Korea for the first time. Their respective 18S rRNA and COI genes were sequenced and used to check p-distances and phylogenetic positions. We also agree that all future description of the genus be from a sound male as first proposed by *Wieser & Hopper (1967)*.

## MATERIAL AND METHODS

### Sampling and morphological study

Three seemingly natural and undisturbed beaches of Jeju Island were sampled in September 11, 2018. Two sub-samples of sediment from the intertidal zone were obtained qualitatively using a mini-shovel. Of the subsamples, one was fixed in 5% neutralized formalin solution for morphological analysis and the other was fixed in 70% ethanol for molecular analysis. Samples were brought back to the laboratory and meiofauna were extracted using the Ludox

method (*Burgess, 2001*). Individual specimens were transferred by hand to a Petri dish filled with 10% glycerin. Specimen-containing Petri dishes were placed for a day in a dry oven preset to 40 °C for a day to achieve complete dehydration as described by *Seinhorst (1959)* with the glycerin-ethanol method. A single specimen was mounted in a drop of glycerin on a slide glass as conferred in the wax-ring method (*Hooper, 1986*). Specimens were examined and identified using Olympus BX51 and Leica DM2500 microscopes. For scanning electron microscopy, specimens were removed from the slide glass and placed in a drop of glycerin. Drops of distilled water were added gradually to the drop of glycerin to rehydrate the specimen. Hydrated specimens underwent ethanol series for dehydration (20%, 40%, 50%, 70%, 80%, 90%, 95%, 100%, for 10 min each) to be placed in hexamethyldisilazane (HMDS), with slightly altered concentration and duration compared to a method used by *Phillips et al. (2016)*. A pool of HMDS containing the specimen was placed in drying oven to be completely dried overnight. Dried specimens were mounted on a stub to be sputter-coated, then observed with a COXEM EM-30 scanning electron microscope.

## DNA extraction and amplification

Each specimen of interest was dissected into head, body, and tail. Heads and tails were retained for morphological analysis and made into permanent slides following *Hooper*'s (*1986*) wax-ring method. The slides were submitted to the National Institute of Biological Resources (NIBR, Korea). Bodies of each specimen were transferred to a well of distilled water for 20 min to be washed of any remaining ethanol. Washed bodies were moved to individual tubes containing 25 μl of worm lysis buffer, prepared prior to extraction following *Williams et al. (1992)*. The tubes were then placed in PCR-thermo cycler (Takara, Japan) preset to 65 °C for 15 min, 95 °C for 20 min, and 15 °C for 2 min. Two gene loci commonly used for marine nematodes were sequenced: mitochondrial cytochrome oxidase C subunit I (COI) gene and 18S small subunit ribosomal rRNA. All genes were amplified using PCR premix (Bioneer Co., Daejeon, Korea) with 5 μl DNA template, 15 μl distilled water, 1 μl of each primer. COI genes were amplified using primer sets (JB3/JB5) amplifying approximately 300 base pairs (bp) as described by *Derycke et al. (2010)*. PCR cycling conditions were: 94 °C for 5 min, 35 cycles of (94 °C for 30 s; 50 °C for 30 s; 72 °C for 30 s), and 72 °C for 10 min. 18S rRNA was amplified using primer sets (MN18F/22R), amplifying approximately 300 bp. PCR cycling conditions were: 95 °C for 5 min, 37 cycles of (95 °C for 30 s, 56 °C for 1 min, 72 °C for 1 min 30 s), followed by 72 °C for 5 min, as described by *Bhadury et al. (2006)*. Success of amplification was determined by electrophoresis on 1% agarose gel. If amplification was successful, DNA templates were sent to Macrogene (Korea), to be sequenced on an ABI3730XL sequencer.

## Molecular data analysis

Sequenced forward and reverse strands were visually checked for signal quality using FinchTV (ver. 1.4.0). Two strands were aligned with ClustalW (*Thompson, Higgins & Gibson, 1994*) implemented into MEGA (ver. 7.0.26) (*Kumar, Stecher & Tamura, 2016*) with default parameters. All aligned sequences were confirmed with BLAST search (*Altschul et al., 1990*) on GenBank to check that the sequences were those of nematodes. Pairwise

distances between mtCOI and 18S rRNA sequences were calculated using the K2P model (*Kimura, 1980*) using MEGA 7. The best fit-model for 18S rRNA datasets were assessed using default parameters implemented in MEGA 7.0 (*Kumar, Stecher & Tamura, 2016*). Tamura 3-parameter model (*Tamura, 1992*) with gamma distribution of rates across sites was found to be optimal and used with MEGA 7.0 to build a maximum likelihood (ML) tree with complete deletion and 1,000 bootstrap repetition. Completed tree was exported to FigTree (ver. 1.4.4) (*Rambaut, 2009*) and visually modified. Phylogenetic tree was not constructed using the obtained mtCOI sequences, as only few mtCOI sequences of *Enoploides* were available on GenBank.

## Bibliographical revision of the genus

The Bremerhaven Checklist of Aquatic Nematodes by *Gerlach & Riemann (1974)* was initially used to collect original descriptions and references. Original erection of the genus, as well as other previous revisions and diagnosis of the genus was checked (*Wieser, 1953*; *Wieser & Hopper, 1967*; *Platt & Warwick, 1983*; *Smol, Muthumbi & Sharma, 2014*). Upon collecting all required references: (1) validity and synonymy of species were examined and determined; (2) a table comparing diagnostic characters of all valid species was compiled; (3) locality and distribution of original descriptions were determined; (4) a new complete key to the genus was compiled.

## Nomenclatural acts

The electronic version of this article in Portable Document Format (PDF) will represent a published work according to the International Commission on Zoological Nomenclature (ICZN), and hence the new names contained in the electronic version are effectively published under that Code from the electronic edition alone. This published work and the nomenclatural acts it contains have been registered in ZooBank, the online registration system for the ICZN. The ZooBank LSIDs (Life Science Identifiers) can be resolved and the associated information viewed through any standard web browser by appending the LSID to the prefix http://zoobank.org/. The LSID for this publication is: urn:lsid:zoobank.org:pub:6F60918D-9DE1-4B75-A251-C01E0694D01F. The online version of this work is archived and available from the following digital repositories: PeerJ, PubMed Central and CLOCKSS.

## RESULTS

### Systematics

Order Enoplida *Filipjev, 1929*
Family Thoracostomopsidae *Filipjev, 1927*
Subfamily Enoplolaiminae *De Coninck, 1965*
Genus Enoploides *Ssaweljev, 1912*

**Generic diagnosis**: (Updated from *Wieser, 1953*; *Wieser & Hopper, 1967*; *Platt & Warwick, 1983*; *Smol, Muthumbi & Sharma, 2014*).

Enoplolaiminae. Lips high and striated. Buccal cavity with three well-developed solid mandibles with claw-like anterior; mandible not extremely slender (ratio length/width <6); Three onchia shorter than the mandibles. Some species showing sexual dimorphism with pilosity either along the body or within the head region. Spicules usually long, some short, armed with either complex s-shaped/simple non s-shaped gubernaculum. Some species with pre-cloacal supplementary organ/papillae or postanal papillae/cuticular element of different form, at varying distances from the cloacal opening. Terminal setae observed at tail tip in some species. Mostly marine, with two freshwater species (*E. fluviatilis* and *E. stewarti*).

Type species: *Enoploides typicus* (*Ssaweljev, 1912*)

**Notes on generic diagnosis:** *Enoploides* can be easily distinguished from other genera such as *Enoplolaimus De Man, 1893* and *Mesacanthion Filipjev, 1927* by the morphology of its mandibles. Mandibles of *Enoploides* are described as ''solid'', signifying that the two lateral bars converge/fuse together to form a single rod for most of its length. This means that from lateral view, the mandible resembles the letter Y, with its distal end of two lateral bars claw-like, curving inward to the lumen (Fig. 1A). *Enoploides* are sometimes mistaken for *Epacanthion Wieser, 1953* (a closely related genus), and vice versa. Mandibles of *Epacanthion* consist of two lateral bars separated by a thin sheet of cuticles, meaning the space between bars is not solid as in *Enoploides* (Fig. 1B). This subtle difference is significant enough to separate the two genera, yet easily missed in many diagnoses. *Greenslade & Nicholas (1991)* transferred three species previously regarded as members of *Epacanthion* to the genus *Enoploides* (*E. crassum*, *E. filicaudatum* and *E. incurvatus*) after examining their mandibles.

### List of valid species

1. **Enoploides amphioxi** (*Filipjev, 1918*: 92–92, Tables 2, 3, fig. 12A–E; one male and several females, Sevastopol, Black Sea, Russia. *Schuurmans Stekhoven Jr, 1950*: 52, figs. 19A–C; one female, Mediterranean, Villefranche, grey mud, 80 m, deep).

2. **Enoploides bisulcus** (*Wieser & Hopper, 1967*: 252–253, figs. 1I, 10A–D; description based on several males and females, Key Biscayne, Florida, USA, shallow water close to submerged patch, fine sand and debris).

3. **Enoploides brunettii** (*Gerlach, 1953*: 527–529, Abb. 4, figs. 4A–4E; description based on two males and one female, Mediterranean. *Warwick, 1971*: 444–451; Exe estuary, England).

4. **Enoploides caspersi** (*Riemann, 1966*: 186–188, Abb. 49A–F; description based on three male and one female, Elbe estuary, North Sea, Germany).

5. **Enoploides cephalophorus** (*Ditlevsen, 1918*); *Filipjev, 1927* (*Ditlevsen, 1918*: 207–208, Pl. 14, figs. 1, 5, 6, Pl. 15, fig. 1; (=*Enoplolaimus cephalophorus*), description based on several males and females, Limfjord, off Snoghøj, off Hellebæk, Denmark. *Filipjev, 1927*: 142; (as *Enoploides cephalophorus*), several males and females, Kara Sea, yellow sand, 20 m deep. (*Allgén, 1946*): 5; (as *Enoplolaimus* (*Enoploides*) *cephalophorus* Ditlevsen), several males and females, Norway).

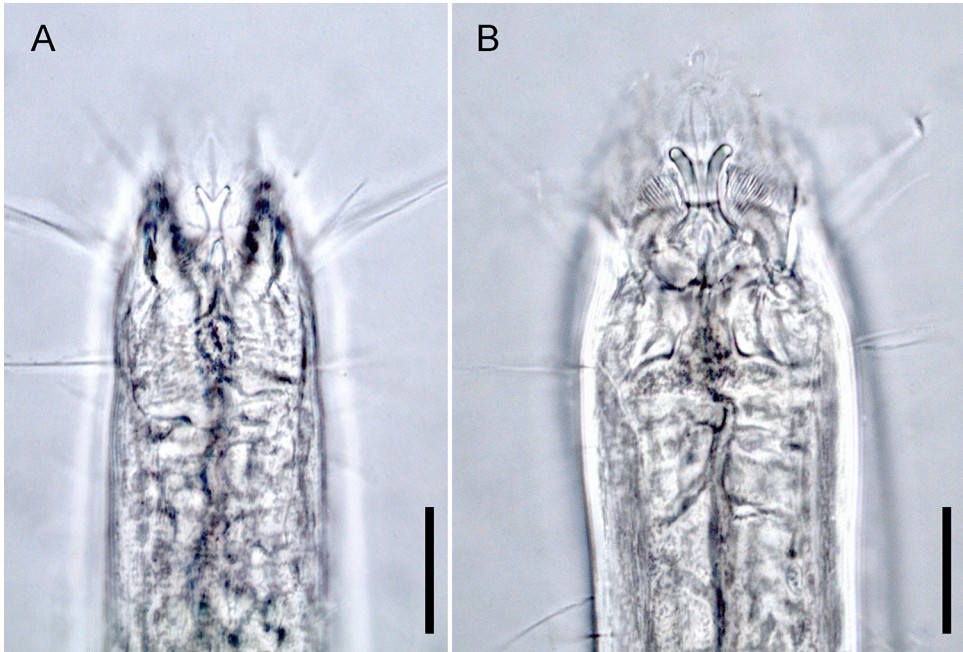

**Figure 1  Head region, showing different type of mandibles.** (A) *Enoploides koreanus* sp. nov., male. (B) *Epacanthion hirsutum* *Shi & Xu, 2016*, male. Scale bars: 20 μm (A and B).

6. ***Enoploides cirrhatus*** (*Filipjev, 1918*: 101–103, Table 3, fig. 15A–D; description based on one male, Shimit Bay, Sevastopol, Black Sea, Russia, saccocirrous sand. *Allgén, 1940*: 263, figs. 1A–1C; (as *Enoplolaimus* (*Enoploides*) *cirrhatus* Filipjev), Norway).

7. ***Enoploides crassum*** (***Ditlevsen, 1926***); ***Greenslade & Nicholas, 1991*** (*Ditlevsen, 1926*: 39–40, Pl. 15, figs 3, 6, 7, 8; (=*Enoplolaimus crassus*), description based on one male and several females, Iceland, Faroe Island, Jan Mayen. *Wieser, 1953*: 79; (=*Epacanthion crassus*), transfers the species to the genus *Epacanthion*. *Greenslade & Nicholas, 1991*: 1041; (=*Enoploides crassus*), transfers the species to the genus *Enoploides*, explaining that the original drawing and description of the mandible does not agree with character of *Epacanthion*. *Smol, Muthumbi & Sharma, 2014*: 202; lapsus *crassum*).

8. ***Enoploides delamarei*** (*Boucher, 1977*: 746–748, figs. 6A–6G; description based on three males, four females and three juveniles, Pierre Noire (Western Channel), France, infralittoral fine sands).

9. ***Enoploides disparilis*** (*Sergeeva, 1974*: 122, figs. 2A–2B; description based on one male, Black Sea, fine sand, 10 m deep).

10. ***Enoploides fluviatilis*** (*Mikoletzky, 1923*: 13–15, figs. 1A–1C; description based on four males, five females and three juveniles, Volga river (freshwater), Russia. Also from brackish-water Caspian Sea (ca 13‰)—data of Tchesunov, A.V.).

11. ***Enoploides gryphus*** (***Wieser & & Hopper, 1967***: 235, Pl. 3, fig 11C, Pl. 5, figs. 11A–B; description based on several males, Virginia Key, Florida, USA).
12. ***Enoploides harpax*** (*Wieser, 1959*: 21–22, fig. 16A–C; description based on one male and one female, Alki Point, Golden Gardens, Richmond Beach, Seattle Washington, USA).

13. ***Enoploides hirsutus*** (*Filipjev, 1918*: 97–100, Table 3, fig 13; description based on one male, Sevasotopol, Black Sea, Russia, mud).

14. (***Enoploides incurvatus***) (***Ditlevsen, 1926***); ***Greenslade & Nicholas, 1991*** (*Ditlevsen, 1926*: 37, Pl. 14, figs. 4, 5, 8, 9, Pl. 15, fig 5; (=*Enoplolaimus incurvatus*), description based on male and female, Hanstholm, Denmark. *Schuurmans Stekhoven Jr, 1946* 44, figs. 13A–O; (=*Enoploides incurvatus*), two males, two females and one juvenile, Langesundsfjord, Skageraks, Sweden, rocky bottom, 150–200 m deep. *Wieser, 1953*: 79; (=*Epacanthion incurvatus*). *Greenslade & Nicholas, 1991*: 1034; mentioned that *Schuurmans Stekhoven Jr*'s (*1946*) redescription was of different species, renames the species *Epacanthion stekhoveni*, on account of a space between the mandibular columns).

15. ***Enoploides labiatus*** (***Bütschli, 1874***); ***Filipjev, 1918*** (*Bütschli, 1874*: 41, figs. 36A–B; (=*Enoplus labiatus*), description based on female, North Sea. *Filipjev, 1918*: 91; (=*Enoploides labiatus*). *Wieser, 1953*: 87; the species is considered as *Enoploides*, but also synonymous with *E. spiculohamatus Schulz, 1932*. *Wieser & Hopper, 1967*: 251; mentions the species as a doubtful species and adds that synonymy of *E. labiatus* and *E. spiculohamatus* cannot be proven and thus should be abandoned. *Bouwman, 1981*): 63, fig26; (=*Enoploides cf labiatus Bütschli, 1874*), 6 males, 10 females and dozens of juveniles, the Ems Estuary, mentions that there is high probability that *E. labiatus* is synonymous with *E. spiculohamatus*).

16. ***Enoploides labrostriatus*** (***Southern, 1914***); ***Filipjev, 1921*** (*Southern, 1914*: 53–54, Pl. 8, figs. 24A–F; (=*Enoplus labrostriatus*), description based on males, females and juveniles, Clew Bay, Ireland, sand and shells, 44 m deep (converted from fathoms). *Filipjev, 1921*): 565–567; transfers the species to the genus *Enoploides*. *Filipjev, 1927*: 141; acknowledges the species as the genus *Enoploides*).

17. ***Enoploides longispiculosus*** (*Vitiello, 1967*: 407–410, fig. 3A–3G; description based on two males, one female and two juveniles, English Channel).

18. ***Enoploides mandibularis*** (*Coles, 1977*: 25–27, figs. 8A–C; description based on seven males and ten females; Saldanha Bay, False Bay, South Africa. The mandible is described as "solid" and with broad central expansion on the side facing the buccal cavity. The original figures on the mandibles raise some questions regarding its placement in the genus. Types will have to be examined for further determination).

19. ***Enoploides polysetosus*** (*Jensen, 1986*: 93–94, figs. 1A–1G; description based on seven males, seven females and thirteen juveniles, East Flower Garden, NW Gulf of Mexico).

20. ***Enoploides ponticus*** (*Sergeeva, 1974*: 122, figs. 3A–3B; description based on one male, Black Sea, silt, 82 m deep).

21. ***Enoploides rimiformis*** (*Pavljuk, 1984*: 1145–1146, figs. 1D–1Z (in Russian alphabet); description based on males and females. Sea of Japan (East Sea), sand).

22. ***Enoploides spiculohamatus*** (*Schulz, 1932*: 341–344, figs. 5A–5K; description based on males and female, Kiel Bay. *Benwell, 1981*: 177–181, figs. 1A–1D; two males and

two juveniles examined, Scotland, states that *Bresslau & Schuurmans Stekhoven (1940)* may be *E. spiculohamatus*, but the description is poor; also states that *Schuurmans Stekhoven Jr (1935)* is not *E. spiculomahatus*, but lacks details to consider it a new species).

23. **Enoploides stewarti** (*Nicholas, 1993*): 167–170, figs. 2E–2F, 3A–E; description based on several males and females, Lake Alexandrina, South Australia, sand at water edge of freshwater).

24. **Enoploides typicus** (*Ssaweljev, 1912*: 115; description based on one male, no depiction, Kolafjord, Russia, mud, 70–80 m deep).

25. **Enoploides tyrrhenicus** (*Brunetti, 1949*: 42–44; Mediterranean. *Gerlach, 1953*: 526–527, figs. 3A–3E; three males, Mediterranean).

26. **Enoploides vectis** (*Gerlach, 1957*: 426, figs. 4C–4G; (=*Enoploides brunettii* var. *vectis* var. n.), description based on a male, Rio de Janeiro, Brazil, middle sand. *Wieser & Hopper, 1967*: 252; raised to species level).

## Species inquirendae

1. *Enoploides brattstroemi* (*Wieser, 1953*: 88–89, figs. 47A–B; description based on two juveniles, Gulf of Corcovado and Boca del Guafo, Chile, littoral and sublittoral, sheltered algae and coarse bottom, 25 m deep, lapsus *brattströmi*. *Wieser & Hopper, 1967*: 251; argues that classification within this genus is only possible using the male genital armature, and classifies any existing descriptions based on only juveniles and females as *species inquirendae*).

2. *Enoploides brevis* (*Filipjev, 1918*: 100–101, Table 3, fig. 14; description based on immature female, Sevastopol, Black Sea, Russia. *Wieser & Hopper, 1967*: 251; argues that classification within this genus is only possible using the male genital armature, and classifies any existing descriptions based on only juveniles and females as *species inquirendae*).

3. *Enoploides filicaudatum* (*Mawson, 1956*); *Greenslade & Nicholas, 1991* (*Mawson, 1956*: 64, figs. 27A–B; (=*Epacanthion Filicaudatum*), description based on two juveniles, Antarctic. *Greenslade & Nicholas, 1991*: 1041, fig. 6; transfers the species to the genus *Enoploides* after examining a juvenile kept in South Australian Museum collection (SAMA V3267), which they designated as lectotype. According to them, the mandibles are distinctly solid with mandibular rods joined medially, fitting of mandibles found in the genus *Enoploides*. This species is placed as species inquirenda due to the following reasons: (1) The original description was based on two juveniles, which *Greenslade & Nicholas (1991)* later could only locate one. We agree with *Wieser & Hopper (1967)* whom insisted descriptions based on only females and juveniles are insufficient to distinguish between species; (2) While *Greenslade & Nicholas (1991)* confirmed that the mandibles of this species resemble ones found in the genus *Enoploides*, it is still difficult to distinguish it from other species within the genus. For instance, its species defining characteristic as according to *Mawson (1956)* is for its tail shape. This characteristic alone is too ambiguous to define a species).

4. *Enoploides italicus* (*Steiner, 1921*; *Filipjev, 1918*) (*Steiner, 1921*: 54, fig. A$^1$; (=*Enoplolaimus italicus*), no locality, no measurements. *Filipjev, 1927*: 141; (=*Enoploides italicus*). *Wieser, 1953*: 88; mentions the species being doubtful, reasoning original description only being provided with a figure of a head with 12 setae without description).

5. *Enoploides kerguelensis* (*Mawson, 1958*: 345, figs. 27A–C; description based on one female and three juveniles, Kerguelen Island, Antarctica. *Wieser & Hopper, 1967*: 252; argues that classification within this genus is only possible using the male genital armature, and classifies any existing descriptions based on only juveniles and females as *species inquirendae*).

6. *Enoploides longicaudatus* (*Wieser, 1953*: 91, figs. 51A–B; description based on two juveniles, Golfo de Ancud, Chile, coarse sand, small stones and a few boulders, much detritus, 30–40 m deep. *Wieser & Hopper, 1967*: 252; argues that classification within this genus is only possible using the male genital armature, and classifies any existing descriptions based on only juveniles and females as *species inquirendae*).

7. *Enoploides longisetosus* (*Schuurmans Stekhoven Jr, 1943*: 338–339, figs. 10A–B; description based on a juvenile, Chile. (*Wieser, 1953*: 88; mentions that the species is doubtful because only one juvenile is known and cervical and body setae are very long. Based on the fact that the description is based on one juvenile, the species is considered species inquirenda).

8. *Enoploides macrochaetus* (*Allgén, 1929*) *De Coninck & Schuurmans Stekhoven, 1933* (species inquirenda) (*Allgén, 1929*: 15–16, figs. 5A–5B; (=*Enoplolaimus macrochaetus*), description based on a juvenile, Skagerrak, Sweden. *Allgén, 1953*: 555; transfers the species to the genus *Enoploides*. *De Coninck & Schuurmans Stekhoven, 1933*: 39; labels it a doubtful species due to the original description lacking sufficient figures).

9. *Enoploides oligochaetus* (*Mawson, 1956*: 67–68, figs. 31–33; description based on three juvenile females, Antarctica, no mud, 163 m deep. *Wieser & Hopper, 1967*: 252; argues that classification within this genus is only possible using the male genital armature, and classifies any existing descriptions based on only juveniles and females as *species inquirendae*, lapsus *oligotricha*).

10. *Enoploides paralabiatus* (*Wieser, 1953*: 89, figs. 49A–C; description based on four juveniles and three females, Seno Reloncavi, Chile, sand and mud with terrestrial plant detritus. *Wieser & Hopper, 1967*: 252; argues that classification within this genus is only possible using the male genital armature, and classifies any existing descriptions based on only juveniles and females as *species inquirendae*).

11. *Enoploides pterognathus* (*Mawson, 1956*; 68–69, figs. 32A–B; description based on juvenile females, Antarctica. *Wieser & Hopper, 1967*: 252; argues that classification within this genus is only possible using the male genital armature, and classifies any existing descriptions based on only juveniles and females as *species inquirendae*).

12. *Enoploides reductus* (*Wieser, 1953*: 91, figs. 50A–B; description based on one juvenile, Golfo Corcovado and Boca del Guafo, Chile, coarse sand with some stones, 25 m deep. *Wieser & Hopper, 1967*: 252; argues that classification within this genus is only possible

using the male genital armature, and classifies any existing descriptions based on only juveniles and females as *species inquirendae*).

13. *Enoploides sabulicola* (*Allgén, 1933*) *Wieser, 1953* (*Allgén, 1933* : 24–25, figs. 8A–8B; (=*Enoplolaimus sabulicola*), description based on one juvenile, Norway. *Wieser, 1953*: 88; transfers the species to the genus *Enoploides*, but lists it as a doubtful species, likely due to the original description being based on a single juvenile and insufficient description).

14. *Enoploides suecicus* *De Coninck & Schuurmans Stekhoven, 1933* (*Allgén, 1929*: 13–14, figs. 3A–3B; (=*Enoplolaimus savaljevi*), description based on one juvenile, Sweden. *De Coninck & Schuurmans Stekhoven, 1933*: 25; transfers the species to the genus *Enoploides*. *Wieser, 1953*: 88; lists *Enoplolaimus saveljevi* as synonymous to *E. balticus* *Schuurmans Stekhoven Jr, 1935*, while listing *E. balticus* as a doubtful species. Also mentions they are most likely juveniles of *E. labiatus*. *De Coninck & Schuurmans Stekhoven, 1933*: 25; nomen novum *Enoploides suecicus* nom. nov. for *Enoplolaimus saveljevi*).

15. *Enoploides tridentatus* (*Ssaweljev, 1912*: 116; description based on one female, no depiction, Kolafjord, Russia. *Wieser & Hopper, 1967*: 252; argues that classification within this genus is only possible using the male genital armature, and classifies any existing descriptions based on only juveniles and females as *species inquirendae*).

## Nomen nudum

1. *Enoploides tyrannis* *Bussau, 1993* (nomen nudum) (*Bussau, 1993*: 465–469, figs. 198A–C, 199A–D; description based on one male and two females. *Bussau, 1993* is considered nomen nudum as it does not conform to Article 13 of *ICZN (1999)*.

2. *Enoploides uniformis* *Pavljuk, 1984* (*Pavljuk, 1984*: 1145–1146, figs. 1D–1Z (in Russian alphabet); (=*Enoploides rimiformis*), description based on males and females. Sea of Japan (East Sea), sand. This species name is an accepted name on NeMys, however, *Pavljuk, 1984* does not include a description of a species with such name. It is likely a genuine mistake confusing the name of *rimiformis* as *uniformis*).

## *Enoploides koreanus* sp. nov.

Figs. 1A, 2, 3, Table 1
urn:lsid:zoobank.org:act:87CC02F7-3A2E-4137-84E5-C8E7A97DA6D2

**Description:** Males (Fig. 2; holotype $n = 1$, paratype $n = 2$). Cuticle smooth above cephalic capsule, strongly striated below cephalic capsule until tail tip (Figs. 4A, 4B). Three lips high with its border striated heavily with grooves, each lip with two inner labial setae. Six inner labial setae, long and thin (11 μm long), at base of lips in one crown. Six longer outer labial setae (43 μm long) and four shorter cephalic setae (15 μm long) in one crown, situated at anterior portion of cephalic capsule. ~20 subcephalic/cervical setae immediately after, some near second crown of setae near outer labial and cephalic setae,

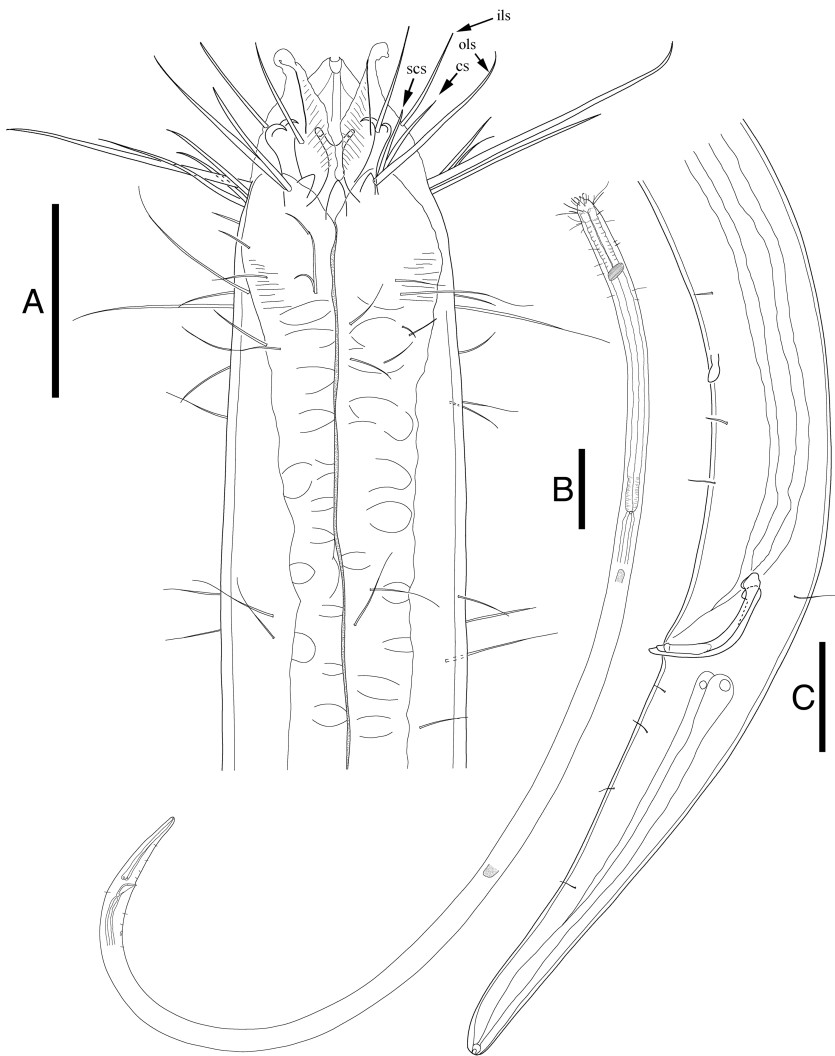

**Figure 2** *Enoploides koreanus* **sp. nov. male.** (A) Head, lateral view. (B) Total view. (C) Tail, with spicules, gubernaculum and preanal supplementary organ. Scale bars: 30 μm (A and C) and 100 μm (B). Figure credit: Raehyuk Jeong.

some near region of cephalic capsule end, in random lengths, some short, some as long as cephalic setae (Fig. 4A). Buccal cavity short and funnel shaped, wide at the anterior end, narrowing gradually towards the base. Buccal cavity armed with three equally sized and shaped "solid" mandibles and teeth. Mandible Y-shaped, two lateral bars converging into one solid bar, with distal end of each lateral bars claw-like, curving inwards to the lumen. Three onchia of equal size, posterior to each base of mandibles. Amphid not observed. Somatic setae irregularly scattered along the cervical region, in random lengths. Pilosity denser from level of buccal cavity until the nerve ring region. Pharynx long with grooves and sinuous external contours. Cardia inverse triangular shaped seemingly embedded into the intestine. Somatic setae sparsely distributed along the body in singles until tail region. Metanemes not observed. Testes paired and opposed, anterior testis slightly right of the

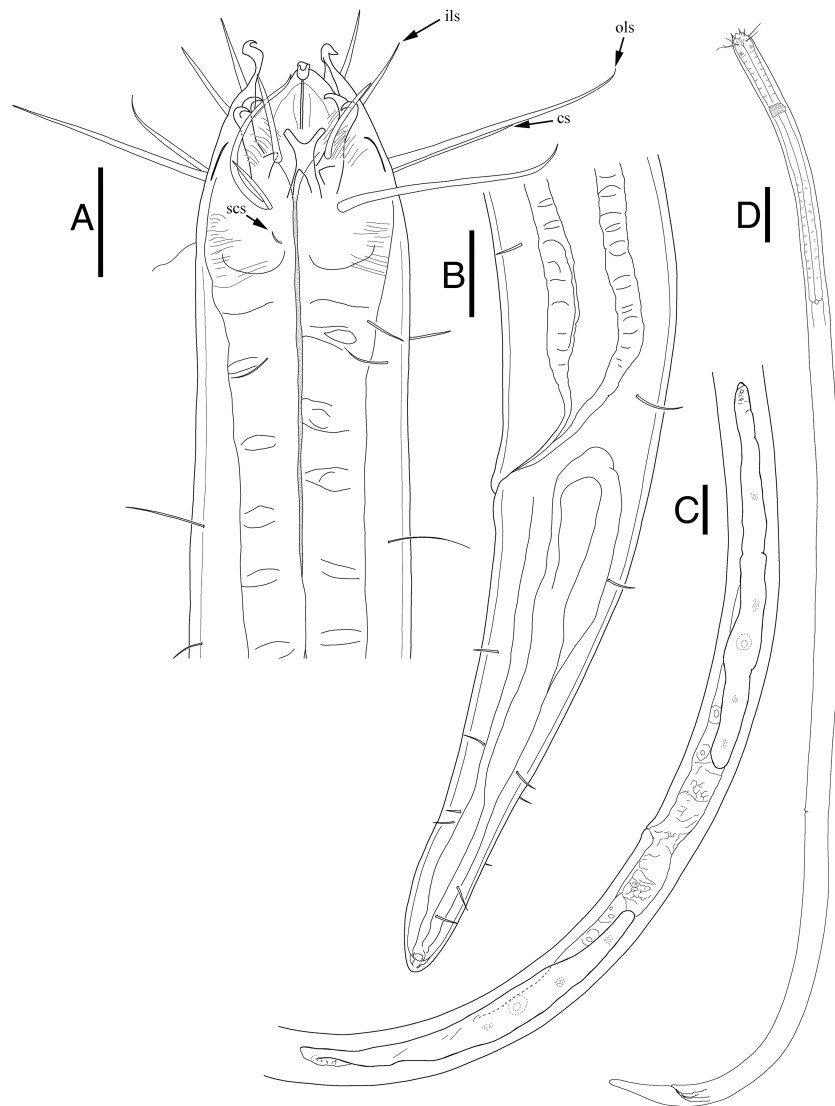

**Figure 3  *Enoploides koreanus.* sp. nov. female.**  (A) Head, lateral view. (B) Tail region with caudal glands. (C) Reproductive system with vulva. (D) Total view. Scale bars: 20 μm (A and B), 40 μm (C) and 100 μm (D). Figure credit: Raehyuk Jeong.

intestine and posterior testis also to right of the intestine. Precloacal supplementary organ, 8 μm long, 82 μm above cloacal opening, roughly 2.6 anal body diameters above the anus. Spicules paired, simple, thin and curved at an obtuse angle, proximal end with a knob (more distinct in some specimens than others) and distal end blunt and rounded. Spicule width equivalent throughout its length. Gubernaculum simple, rod-like shape, running parallel to distal half of spicule from level of spicule curvature to its distal end. Distal end with slightly rounded head (Fig. 4C). Tail region with some somatic setae in singles with no patterns observed. Tail conical, inconspicuously cylindrical at distal end. Caudal glands just below distal end of the spicules, at level of anus, running until a well-developed spinneret.

**Table 1 Measurement of major morphological characters of *Enoploides koreanus* sp. nov.** Measurements are in μm where applicable, and morphometric values are rounded.

| Characters | ♂ holotype | ♂ (*n* = 3) mean ± sd (range) | ♀ (*n* = 6) mean ± sd (range) |
|---|---|---|---|
| Body length | 2,107 | 2,209 ± 100 (2,107–2,307) | 2,136 ± 159 (1,851–2,292) |
| Maximum body diameter | 36 | 35 ± 2 (33–36) | 42 ± 3 (39–47) |
| Diameter at the level of cephalic setae | 29 | 29 ± 1 (29–30) | 29 ± 6 (18–34) |
| Length of inner labial setae | 11 | 14 ± 4 (11–18) | 17 ± 1 (16–19) |
| Length of outer labial setae | 43 | 37 ± 7 (30–43) | 42 ± 4 (39–48) |
| Length of cephalic setae | 15 | 15 ± 4 (11–18) | 17 ± 3 (13–20) |
| Distance from anterior to cephalic setae | 13 | 18 ± 5 (13–23) | 23 ± 3 (19–27) |
| Width at cephalic capsule end | 34 | 34 ± 0 (34–34) | 37 ± 2 (32–38) |
| Mandible length | 12 | 11 ± 1 (10–12) | 12 ± 1 (10–13) |
| Tooth length | 5 | 5 ± 0 (5–5) | 6 ± 1 (5–6) |
| Buccal cavity length | 18 | 24 ± 6 (18–29) | 29 ± 2 (26–30) |
| Distance from nerve ring from anterior end | 127 | 127 ± 10 (117–137) | 133 ± 9 (119–143) |
| Pharynx (oesophagus) length | 510 | 527 ± 17 (510–543) | 524 ± 30 (488–558) |
| Corresponding body diameter at pharynx | 36 | 35 ± 2 (33–36) | 40 ± 2 (36–42) |
| Cardia length | 13 | 10 ± 3 (8–13) | 9 ± 2 (7–12) |
| Tail length | 130 | 125 ± 5 (121–130) | 114 ± 8 (100–122) |
| Anal body diameter | 31.0 | 31 ± 2 (29–33) | 33 ± 3 (29–36) |
| c' | 4 | 4 ± 0 (3.7–4.2) | 3.5 ± 0 (3.3–3.8) |
| Spicule length as arc | 39.0 | 36 ± 3 (34–39) | n/a |
| Spicule length as arc / anal body diameter | 1 | 1.2 ± 0 (1.1–1.3) | n/a |
| Length of gubernaculum | 12 | 12 ± 1 (11–12) | n/a |
| Supplementary organ length | 8 | 9 ± 1 (8–9) | n/a |
| Distance from cloacal opening to supplementary organ | 82.0 | 85 ± 3 (82–87) | n/a |
| Distance from anterior end to vulva | n/a | n/a | 1,355 ± 111 (1,170–1,456) |
| Corresponding body diameter at vulva | n/a | n/a | 42 ± 3 (39–47) |
| Distance from anterior end to vulva as percentage of total body length | n/a | n/a | 63 ± 2 (60–64) |
| a | 58.5 | 63.8 ± 4.6 (58.5–67.1) | 50.6 ± 5 (47.5–58.8) |
| b | 4.1 | 4.2 ± 0.2 (4.1–4.4) | 4.1 ± 0 (3.8–4.3) |
| c | 16.2 | 17.8 ± 1.4 (16.2–18.8) | 18.7 ± 1 (17.5–20.4) |

Several caudal setae observed (Fig. 4B) along the tail with no terminal setae present at tail tip.

Female (Fig. 3; allotype *n* = 1, paratype *n* = 5). Female generally longer and larger in size. Short sub-cephalic setae below outer labial and cephalic setae (Fig. 4D). Cervical setae in singles and less frequent compared to male at posterior end of cephalic capsule. Reproductive system didelphic-amphidelphic, both ovaries reflexed, positioned left of intestine (Fig. 3C). Tail region with some caudal setae in singles with no visible patterns. No terminal setae observed at tail tip.
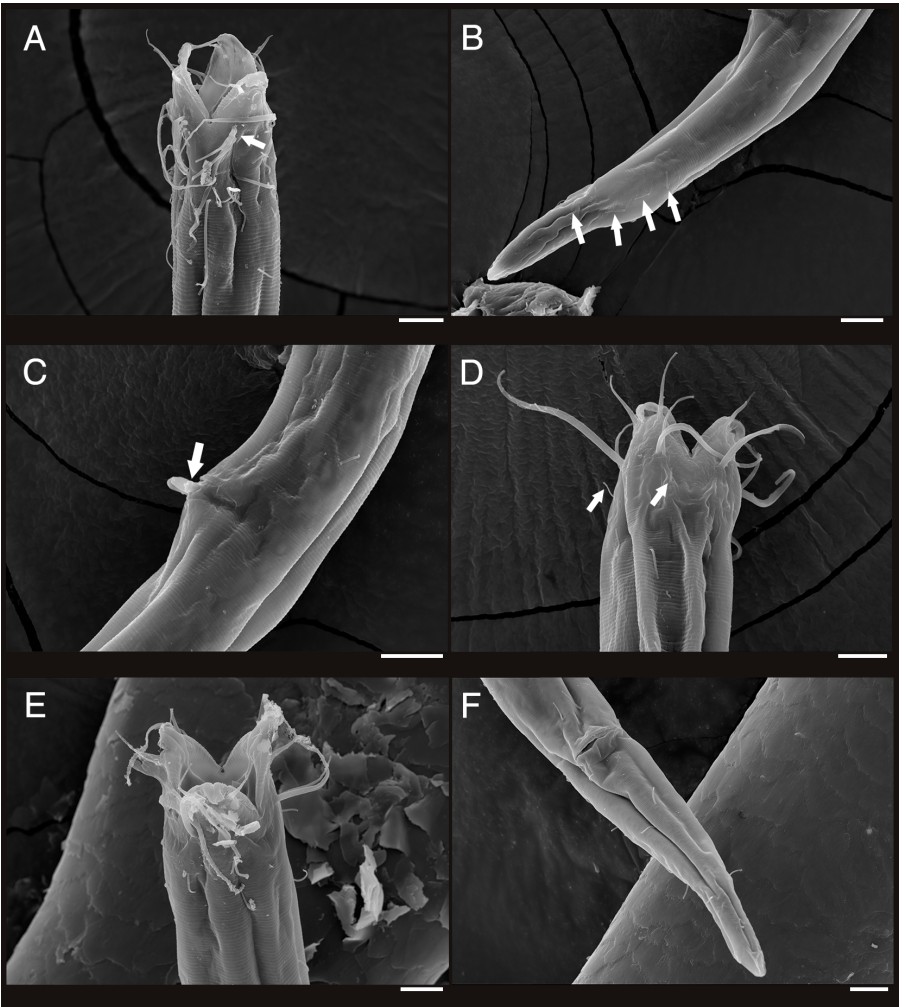

**Figure 4** **Scanning electron micrograph of *Enoploides koreanus* sp. nov.** (A) Male, head region, lateral view, many subcephalic setae on head region tangled to one another. (B) Male, tail region, arrows showing several caudal setae. (C) Male, cloacal opening with arrow showing round head of gubernaculum peeking out, behind it two spicules extending outwards. (D) Female, head region, lateral view, with arrows showing short subcephalic setae below outer labial and cephalic setae. (E) Female, mouth region, subapical view, showing lip configuration. (F) Female, cloacal region, ventral view. Scale bars: 10 μm.

**Type locality:** Intertidal zone at coast of Shinyang Seopjikoji Beach (West), Jeju Island, South Korea (33°26′05″N 126°55′15″E), in sandy beach with some algae, collected 11 September 2018.

**Materials examined:** All specimens deposited in the National Institute of Biological Resources (South Korea). Holotype 1♂ (NIBRIV0000858255) on one slide, Allotype 1♀ (NIBRIV0000858256) on one slide, Paratypes 1♂, 4♀ ♀ all on separate slide (NIBRIV0000858257–NIBRIV0000858261), 1♂ and 2♀ ♀ dried, mounted on two separate stubs, each sex on separate stubs and coated with gold for SEM (NIBRIV0000858262, NIBRIV0000858263).

**Additional materials examined:** All specimens deposited in the National Institute of Biological Resources (South Korea). 1♀, and 3 juveniles all on separate slide but with only head and tails retained, with segment of body used for molecular analysis. (NIBRIV0000858264–NIBRIV0000858267).

**Diagnosis:** *Enoploides.* Male. Body length 1,851–2,307 μm. Cuticle finely striated along the body, smooth only in cephalic capsule region. Six inner labial setae 11–19 μm. Six longer outer labial setae 30–48 μm, four shorter cephalic setae 11–20 μm long sharing one crown. Many subcephalic and cervical setae in head region in case of males, less dense in case of females. Buccal cavity 18–30 μm long. Males with testis paired and opposed. Spicule short (34–39 μm), thin, curved at obtuse angle, its width equal throughout the length. Distal end rounded with knob on proximal end. Gubernaculum simple rod-shaped, running parallel to distal end of spicules. Distal end with subtle round head. Precloacal supplementary organ present. Tail conical, inconspicuously cylindrical at distal end. $a = 58.5$–$67.1$, $b = 4.1$–$4.4$, $c = 16.2$–$18.8$, c'=3.7–4.2.

**Differential diagnosis:** This diverse genus consisting of 27 valid species can be divided into two groups by the length of their spicules: (1) a group with short spicules (<150 μm); (2) a group with long spicules (>150 μm). Instead of comparing just the spicule length, it would have been ideal to separate the group by spicule length/abd. Many species unfortunately lack measurement of anal body diameter in their description, making this a difficult task. Total of seven species have spicules shorter than 150 μm: *E. caspersi*, *E. cirrhatus*, *E. disparilis*, *E. fluviatilis*, *E. stewarti*, *E. tyrrhenicus*, and *E. koreanus* sp. nov. When *Nicholas (1993)* first acknowledged this group and created a key for *Enoploides* with short spicules, he included *E. polysetosus*, but even then it was distinguished for having the longest spicules in the group. We defined this group to consist of those bearing spicules shorter than 150 μm. *E. polysetosus* is therefore no longer considered to have short spicules. Species such as *E. caspersi* and *E. tyrrhenicus* can easily be distinguished from the other by the former's unique post-anal organ and the latter's complex gubernaculum. Of the seven species, the new species is most closely related to *E. disparilis* in terms of general morphology: (1) they both have simple and short spicules (34–39 μm vs. 35 μm) with a knob on its proximal end; (2) they both have simple and short gubernaculum (12 μm vs. 19 μm) parallel to distal end of the spicule; and (3) they share similar body lengths (2,107–2,307 μm vs. 2,250 μm) and index value of b (4.1–4.4 vs. 4.4). They can be differentiated from one another by the following characteristics: (1) the new species has a pre-anal supplementary organ, while *E. disparilis* does not; (2) the new species has a shorter conical tail (121–130 μm), while *E. disparilis* has a longer conico-cylindrical tail (254 μm); (3) the new species and *E. disparilis* have different index values of a (58.5–67.1 vs. 44.7) and c (16.2–18.8 vs. 9.8). Another species similar to the new species is *E. cirrhatus*. which also bears a small spicule with a knobbed proximal end and similarly shaped gubernaculum (adjacent to the spicules, small and curved at upper end). Aside from those characteristics, the new species differ from *E. cirrhatus* in the body lengths (2,107–2,307 μm vs. 4,350 μm), index values ($a = 58.5$–$67.1$ vs. 26; $b = 4.1$–$4.4$ vs. 5.25) all differ greatly, and *E. cirrhatus* does not have a pre-anal supplementary organ, but bear seven pairs of pre-anal papillae instead.

Group of species with long spicules (>150 μm) can be further divided into subgroups by the morphology of the gubernaculum: (1) S-shaped gubernaculum; (2) complex gubernaculum; (3) gubernacula that are short, simple, small, plated, arcuate, or weak. The term "S-shaped" has been used by several authors, including *Filipjev (1918)*, *Wieser & Hopper (1967)*, and *Pavljuk (1984)*, to describe certain shapes of the gubernaculum. *Wieser & Hopper (1967)* conveniently grouped species with this S-shaped gubernaculum (*E. cephalophorus*, *E. gryphus*, *E. spiculohamatus*, *E. amphioxi*, *E. labrostriatus* and *E. bisulcus*) and provided a figure showing different gubernacula of several species within the genus. The S-shape can be extremely general and almost any shape can be regarded as S-shapes given the curving nature of gubernaculum's contours. For instance, gubernaculum of *E. spiculohamatus* was considered S-shaped by *Wieser & Hopper (1967)*, while gubernaculum of *E. vectis* was not. *Wieser & Hopper (1967)* even referred to this group as "more or less S-shaped" confirming its ambiguous nature. To reduce ambiguity, any gubernaculum described as complex in the original description or consisting of multiple parts is not considered S-shaped here. There is certainly some uniformity of morphology of these "complex" gubernacula, which consist of multiple parts, bluntly shaped, anterior to the spicules. This removes *E. spiculohamatus* from the S-shaped group initially assigned by *Wieser & Hopper (1967)*, as its gubernaculum consists of multiple parts and more closely resembles other complex-gubernaculum bearing species such as *E. brunettii*, *E. labiatus*, *E. longispiculosus*, and *E. vectis*. Refer to Table 2 for comparison of diagnostic morphological characters and gubernaculum type of all valid *Enoploides* species.

**Etymology:** The species name refers to its occurrence in Korea.

## Key to all valid species of the genus *Enoploides*

1. Spicules less than 150 μm long…2
   -Spicules more than 150 μm long…7
2. Presence of a prominent post-anal supplementary organ …*E. caspersi*
   -Absence of a prominent post-anal supplementary organ …3
3. Pre-anal supplementary organ absent …4
   -One pre-anal supplementary organ present …5
   -Seven to eight pre-anal supplementary papillae in midline …*E. cirrhatus*
4. index c~9, spicules ~35 μm long and gubernaculum without apophysis …*E. disparilis*
   -index c~12–16, spicules ~90 μm long and gubernaculum with weak apophysis…*E. tyrrhenicus*
5. Tail shorter than 100 μm with two post-anal papillae …*E. fluviatilis*
   -Tail longer than 100 μm …6
6. Spicule ~100 μm long with plate-like gubernaculum with weak apophysis and three terminal setae at tail tip …*E. stewarti*
   -Spicule ~30–40 μm long with rod-like gubernaculum with a rounded head at distal end and no terminal setae at tail tip …*E. koreanus* sp. nov.

7. Gubernaculum S-shaped …16

  -Gubernaculum not S-shaped …8

8. Gubernaculum complex with multiple parts …9

  -Gubernaculum short/small, weak, arcuate, plate …12

9. Pre-anal supplementary organ less than 1 abd away from cloacal opening …*E. vectis*

  -Pre-anal supplementary organ 1.5–1.7 abd away from cloacal opening …*E. spiculo-hamatus*

  -Pre-anal supplementary organ more than 2 abd away from cloacal opening …10

10. Spicule length less than 200 (<4 abd) …*E. brunettii*

   -Spicule length greater than 200 (>4 abd) …11

11. Post-anal cuticular element characteristically S-curved …*E. labiatus*

   -Post-anal cuticular element not s-curved …*E. longispiculosus*

12. Cephalic setae shorter than 10 µm and buccal cavity extremely short (9 µm) …*E. typicus*

   -Cephalic setae longer than 20 µm and buccal cavity average …13

13. Mandible with unique broad central expansion facing the buccal cavity …*E. mandibularis*

   -Mandible generic in characteristic to the genus …14

14. Proximal end of spicule funnel shaped …*E. incurvatus*

   -Proximal end of spicule not funnel shaped, normally curved …15

15. Gubernaculum embracing the distal end of spicule, forming a short plate on each side …*E. crassum*

   -Gubernaculum triangular, unpaired, flat, covering the spicules above and below …*E. hirsutum*

   -Gubernaculum arcuate with wide distal end and narrow and curved proximal end …*E ponticus*

16. Head globular …*E. cephalophorus*

   -Head non-globular …17

17. Presence of cracks on mandibles …*E. rimiformis*

   -Absence of cracks on mandibles …18

18. Dorsal tooth missing …*E. delamarei*

   -Dorsal tooth present …19

19. Distal end of spicule with mobile spine …*E. amphioxi*

   -Distal end of spicule without mobile spine …20

20. Gubernaculum with characteristic ventral knob …*E. gryphus*

   -Gubernaculum without characteristic ventral knob …21

21. Spicule with diagonal reinforcement …22

   -Spicule without diagonal reinforcement …23

22. Spicule longer than 400 µm …*E. bisulcus*

   -Spicule shorter than 300 µm …*E. harpax*

23. Spicule ~490 µm long and smooth, proximal end funnel shaped and distal end slightly expanded and pointed, index c ~20 …*E. labrostriatus*

   -Spicule ~160–170 µm long, almost straight, index c ~11–16 …*E. polysetosus*

**Order Enoplida** *Filipjev, 1929*
**Family Thoracostomopsidae** *Filipjev, 1927*
**Subfamily Enoplolaiminae** *De Coninck, 1965*
**Genus** *Epacanthion Wieser, 1953*
***Epacanthion hirsutum** Shi & Xu, 2016*
**Fig. 5**, **Table 3**

**Description:** Male (Fig. 5; $n = 1$). Cuticle smooth. Lips high with heavy striation and grooves, each lip bearing two inner labial setae. Six inner labial setae, fairly long and thin at base of lips in one crown. Six longer outer labial setae and four shorter cephalic setae in one crown. Cervical setae scattered randomly at posterior end of cephalic capsule, as long as cephalic setae. Buccal cavity funnel shaped, wide at the anterior end, gradually narrowing towards the base. Buccal cavity armed with three equally sized and shaped mandibles and teeth, respectively. Mandibles with two lateral bars diverging away from one another distally. Distal end of each lateral bars "claw-like", curving towards the lumen like hooks. Mandibles widening near the base, each armed with fairly weak, narrow looking onchia. Mandibular columns divided by a sheet of cuticle. Pharyngeal glands not readily visible. Pharynx fairly long and muscular, its width consistent throughout its length, except the swollen anterior end. Fairly long somatic setae in singles, randomly distributed along the head. Pilosity intense until level of nerve ring, scarcer throughout. A little below the level of nerve ring, a ring of densely arranged cervical setae. Cardia inverse triangular, seemingly embedded in the intestine. Testes paired and opposed, both ends positioned left of intestine. Spicules slightly curved with small gubernaculum at distal end of spicules. No precloacal supplementary organ observed. Caudal glands after cloacal opening, well-developed. Tail conical-cylindrical, two long sub-terminal setae observed and two terminal setae at tail tip.

Type species: *Epacanthion buetschlii* (*Southern, 1914*) (28 valid species) (Refer to *Shi & Xu, 2016*).

**Locality:** Intertidal zone at coast of Shinyang Seopjikoji Beach (East), Jeju Island, South Korea (33°26′09.6″N 126°55′29.3″E), in rocky with dark coarse sandy beach, collected 11 September 2018.

**Materials examined:** All specimens deposited in the National Institute of Biological Resources (South Korea). 1♂ (NIBRIV0000858270).

**Additional materials examined:** All specimens deposited in the National Institute of Biological Resources (South Korea). 1♀, and 1 juvenile all on a separate slide but with only head and tails retained, with segment of body used for molecular analysis. (NIBRIV0000858271, NIBRIV0000858272).

**Remarks:** The morphology agrees well to the description provided by the original authors, *Shi & Xu (2016)*. Mandibles clearly resemble those seen in *Epacanthion* species, consisting of two lateral bars (parallel to one another and the space in between not solid) separated by a thin sheet of cuticle. Its distinguishing characteristic, a single row or ring of densely arranged setae below the level of the nerve ring is quite distinct. All measurements are also within the range of the original (Table 3).

Jeong et al. (2020), *PeerJ*, DOI 10.7717/peerj.9037

**Table 2  Comparison of diagnostic morphological characters of all *Enoploides* species.** Males only, morphometric values rounded. Species with spicules shorter than 150 μm marked with asterisk.

| Species | Body length [μm] | a | b | c | c' | Length of Setae | | Spicule length (μm) (spicule length as arc/abd) left/ right if applicable | Gubernaculum (length (μm)) | Gubernaculum type | Supplementary organ/papilla distance from cloacal opening (μm) (supplementary organ distance from cloacal opening/abd) |
|---|---|---|---|---|---|---|---|---|---|---|---|
| | | | | | | Inner labial Setae | Outer labial setae/ cephalic setae | | | | |
| *Enoploides amphioxi Filipjev, 1918* | 5,400 | 51 | 5.5 | 26 | 4.2 | 24 | 50/24 | 500 (12.5) | present (65) | S-shaped | present (3.5) |
| *Enoploides bisulcus Wieser & Hopper, 1967* | 3,500–4,200 | Not measured | Not measured | ~20–21 calc | 4 | 25 | 45–50/25–28 | 420–475 | present (not measured) | S-shaped | present 120–150 |
| *Enoploides brunettii Gerlach, 1953* | 2,060–2,045 | 29–36 | 4.5–4.6 | 14.7–16 | 3–3.5 | 18–24 | 45–53 | 190 (3–3.5) | present (36–39) | Complex with multiple parts | present (1.5–2) |
| *Enoploides caspersi Riemann, 1966** | 3,900–5,150 | 54–60 | 5.4 | 21.4 | Not measured | 27–31.5 | 90/38 | 42.5–48 (0.7 calc) | present (62–72) | S-shaped | present 142–148, post-anal supple-mentary organ also present |
| *Enoploides cephalophorus Ditlevsen, 1918; Filipjev, 1927* | 3,100 | 39 | 4.8 | 17.9 | Not measured | Not measured | Not measured | 264 | present | S-shaped | present 128 |
| *Enoploides cirrhatus Filipjev, 1918** | 4,350 | 26 | 5.25 | 16 | 4.5 | 13 | 40 | 60 (1) | present (15) | Small, adjacent to spicule on dorsal side, upper edge curved | not men-tioned, 7–8 pre-anal papillae present |
| *Enoploides crassum Ditlevsen, 1926; Greenslade & Nicholas, 1991* | 3,000 | 21 | 3.9 | 17 | Not measured | Not measured | Not measured | "very long" | present | Short plates on distal end of spicule | present (at level of middle of spicule) |
| *Enoploides delamarei Boucher, 1977* | 2,090–2,693 | 37.3–43.1 | 4.5–5.3 | 14.8–21.4 | 3.4–4.6 calc | 18–19 | 45–47/21 | 413–568 (10–19.6 calc) | present | S-shaped | present (3.9–4.6) |
| *Enoploides disparilis Sergeeva, 1974** | 2,250 | 44.7 | 4.4 | 9.8 | Not measured | 12.5 | Not measured | 35 (0.9) | present (19) | Small | absent |
| *Enoploides fluviatilis Miko-letzky, 1923** | 1,340–1,900 | 42.5–53 | 2.4–3.23 | 25–31 | Not measured | 8.5–10 | 24–27/10 | (1.46–1.84) | present | Unclear | present (2) |
| *Enoploides gryphus Wieser & Hopper, 1967* | 3,000–3,700 | | ~3.7–4.1 calc | ~15 calc | ~4 calc | 28–32 | 60–80/20–27 | 230–260 (~4.3–5.1 calc) | present | S-shaped | present 80–110 (~1.8 calc) |
| *Enoploides harpax Wieser, 1959* | 4,150 | 50 | 4.4 | 27.7 | 3.5–4 | 25 | 62/28 | 290 (5.8 calc) | present | S-shaped, with two hooks | present 120 (2.4 calc) |
| *Enoploides hirsutus Filipjev, 1918* | 2,850 | 24 | 3.8 | 19 | Not measured | 8 | 27/23 | 425 (8.3) | present (43) | Unpaired, flat triangular | present (3) |
| *Enoploides incurvatus Ditlevsen, 1926; Greenslade & Nicholas, 1991* | 4,300 | 30 | 5 | 18.7 | Not measured | Not measured | Not measured | 176 *measured in linear | uncertain "small" | Small | present 160 |
| *Enoploides labiatus Bütschli, 1874; Filipjev, 1918* | 2,800–3,700 | 22–29 | 4.1–4.7 | 17–20.5 | Not measured | 0.25 head diameter | 1 head diameter/ 0.5–0.6 head diameter | 300–330 (5.0–5.9) | present (1 abd) | Complex with multiple parts | present (3) |
| *Enoploides labrostriatus Southern, 1914; Filipjev, 1921* | 5,550 | 42 | 6.85 | 20 | 4.6 calc | Not measured | Not measured | 490 (8.1 calc) | present | S-shaped "tubu-lar" | present (~2.3 calc) |
**Table 2** (*continued*)

| Species | Body length [μm] | a | b | c | c' | Length of Setae | | Spicule length (μm) (spicule length as arc/abd) left/ right if applicable | Gubernaculum (length (μm)) | Gubernaculum type | Supplementary organ/papilla distance from cloacal opening (μm) (supplementary organ distance from cloacal opening/abd) |
|---|---|---|---|---|---|---|---|---|---|---|---|
| | | | | | | Inner labial Setae | Outer labial setae/ cephalic setae | | | | |
| *Enoploides longispiculosus* Vitiello, 1967 | 3,700– 4,055 | 20.9–22.1 | 4.4 | 17.6–19.4 | Not measured | 15–18 | 51–64/42–43 | 460 (5.4) | present (74) | Complex, massive, strongly cu- ticularized with multiple parts | present 233 (2.7) |
| *Enoploides mandibularis* Coles, 1977 | 4,800– 6,400 | ~24–26.6 calc | ~4.8–5.8 calc | ~15–16 calc | ~3.5–4 calc | 10 | 30/20 | 400–460 (~4.4–4.6 calc) | present | "short, weakly developed" | present 350 |
| *Enoploides polysetosus* Jensen, 1986 | 4,380– 5,160 | 41–60 | 3.8–4.5 | 11.7–15.8 | 5.4 calc | 20 | 45–50/28–30 | 162–174 (~2.8 calc) | present | S-shape with two teeth | present 132–158 (~2.3 calc) |
| *Enoploides ponticus* Sergeeva, 1974 | 3,345 | 21.3 | 3.4 | 19.3 | 2.7 calc | 21 | 23 | 343–363 | present (54) | Arcuate, con- sist of two parts: distal wide and proximal narrow/curved | present 175 (2.7) |
| *Enoploides rimiformis* Pavljuk, 1984 | 2,200– 2,300 | 44–55 | 4.6–5 | 19–20.3 | | 20 | 43/21 | 188–192 (~7) | present (33) | S-shaped with 2 hooks | present 75 (3) |
| *Enoploides spiculohamatus* Schulz, 1932 | 2,340– 3,030 | 28.2–34.3 calc | 4.1–4.6 calc | 15.1–15.9 | 2.7–3 | 20 | 48/24–25 | 280–356 | present (42–45) | Complex struc- ture with multi- ple parts; pair of plates joined by process at prox- imal end, distal end grooved with claw-shaped pro- jection on dor- sal groove, two rounded projec- tions in ventro- lateral position. | present 89–94 (1.5–1.7 calc) |
| *Enoploides stewarti* Nicholas, 1993[*] | 1,930– 3,080 | 28–49 | 3.9–4.9 | 17–24 | 1.7–2.7 calc | 8–15 | 27–37/14–19 | 107–121 (2.1–2.6) | present | Simple plate with very weak apoph- ysis | present 78–118 (1.5–2.4 calc) |
| *Enoploides typicus* Ssaweljev, 1912 | 2,600 | 40 | 4.6 | 14 | Not measured | Not measured | 8.2 | Not measured | present | Similar to *Mesacanthion tenuicaudatus* "median plate with two lateral grooves with loops" | unclear |
| *Enoploides tyrrhenicus* Brunetti, 1949[*] | 1,776– 2,044 | 30–40 | 3.3–3.6 | 12.2–16.3 | | 24 | 70–75/30 | 87–90 | present | "Complicated" | absent |
| *Enoploides vectis* Gerlach, 1957 | 3,160 | 50 | 4.2 | 15.8 | 3.8 calc | 27 | 64/20 | 340 | present (57) | Complex with bent projection at proximal end | present 41 (0.8) |
| *Enoploides koreanus* sp. nov.[*] | 2,107– 2,307 | 58.5–67.1 | 4.1–4.4 | 16.2–18.8 | 3.7–4.2 | 11–18 | 30–43/11–18 | 34–39 (1.1–1.3) | present | Rod-like plate with round head on distal end | present 82–87 |

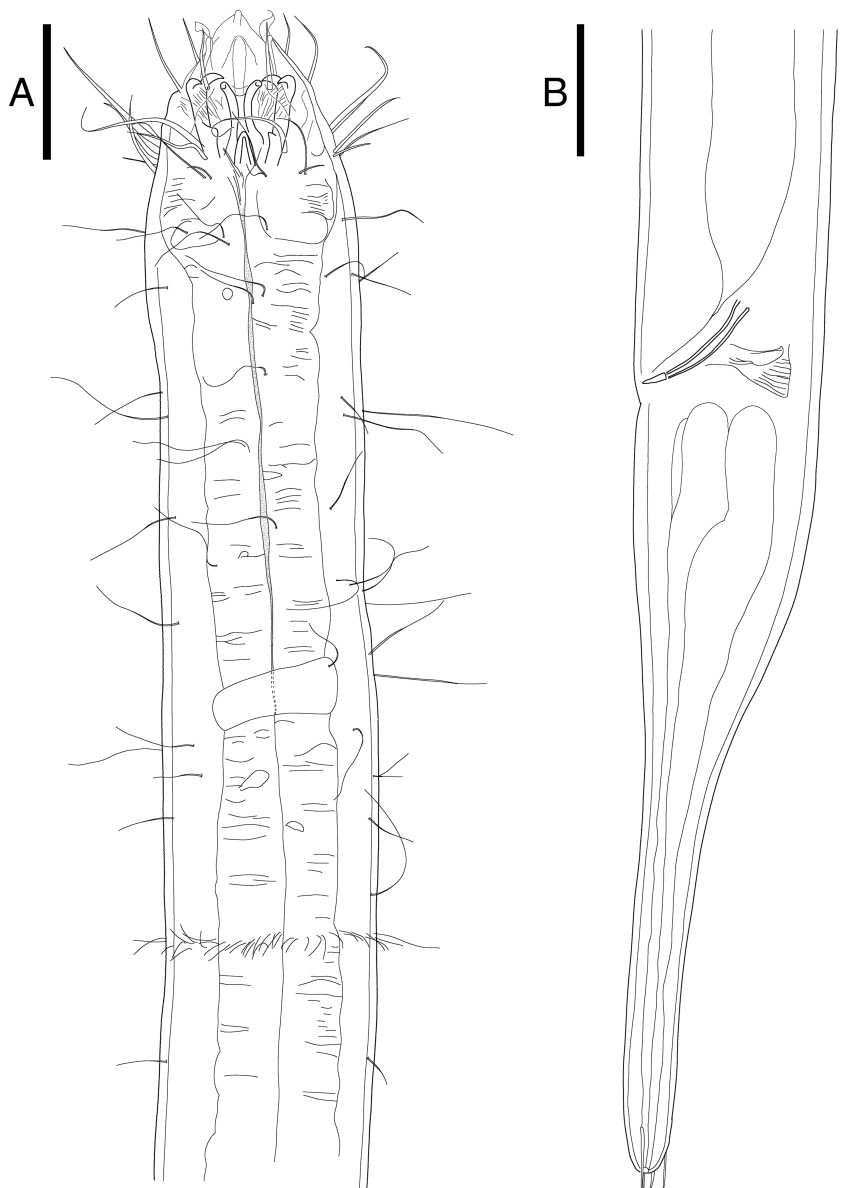

**Figure 5** *Epacanthion hirsutum Shi & Xu, 2016*. **male.** (A) Head, lateral view. (B) Tail region with spicules and gubernaculum. Scale bars: 30 μm (A and B). Figure credit: Raehyuk Jeong.

## Molecular analysis
### Mitochondrial cytochrome oxidase C subunit 1 (mtCOI)

We successfully amplified and sequenced DNA of four *Enoploides koreanus* sp. nov. and two *Epacanthion hirsutum*. Despite the JB3/JB5 primers being commonly used in molecular studies of nematodes (*Derycke et al., 2005*; *Derycke et al., 2010*; *Derycke et al., 2016*; *Avó et al., 2017*) few *Enoploides* sequences were available on GenBank to produce a meaningful phylogenetic analysis. Instead, the pairwise distance of all available *Enoploides* mtCOI sequences was calculated using K2P-substitution model using MEGA 7.0. There was no

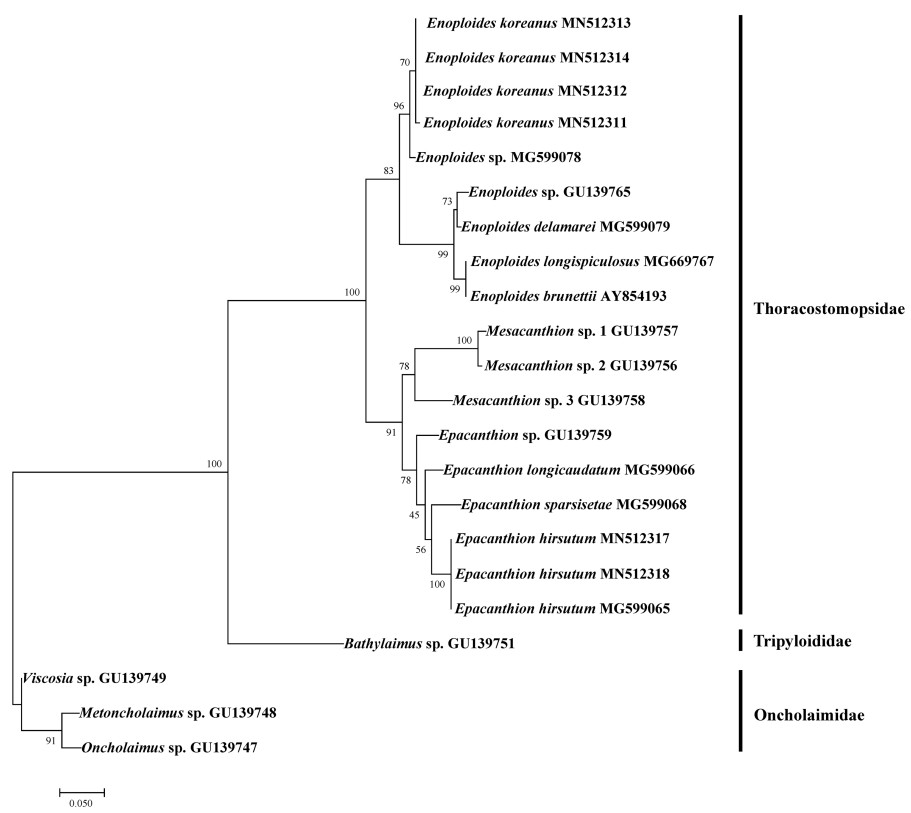

**Figure 6** Maximum likelihood tree of 18S rRNA gene based on TPM3+G (with bootstrap values shown at each nodes; 1,000 replicates).

genetic divergence between the new species, while in comparison to other congeners, 19% to 24% divergence was seen (Table 4). This is well within range to genetic divergence seen between congeners using mtCOI sequences (*Derycke et al., 2010*).

## 18s rRNA

We successfully amplified and sequenced DNA of four *Enoploides koreanus* sp. nov. and two *Epacanthion hirsutum*. To test how our sequences group with existing sequences on GenBank, we rebuilt 18S rRNA ML tree by *Pereira et al. (2010)* with most *Enoploides* 18S sequences available on GenBank (listed in Table 5). The reason for selecting their tree was because their study dealt with a number of free-living marine nematodes (especially Thoracostomopsidae) and used the same primer sets utilized in the present study. The ML tree was more or less similar to the one provided by *Pereira et al. (2010)*, with Thoracostomopsidae forming monophyletic clade with 100% bootstrap. *Enoploides koreanus* and *Epacanthion hirsutum* sequences obtained from this study both formed a clade with their respective congener species, with 83% and 78% bootstraps respectively (Fig. 6). One 18S rRNA sequence of *Epacanthion hirsutum* (MG599065) on GenBank was also retrieved to examine similarities. Although this sequence is considerably longer than ours (1,671 bp vs. 310–311 bp), for regions which do overlap, they showed no differences at any of the sites. Sequences obtained in this study have been submitted to GenBank and

**Table 3 Comparison of diagnostic morphological characters between *Epacanthion hirsutum* found from Korea and from original description.** Males only, morphometric values rounded.

| Characters | ♂ (*n* = 1) from Jeju Island, Korea | ♂ (*n* = 10) from original description (*Shi & Xu, 2016*) |
| --- | --- | --- |
| Body length | 2,468 | 1,879–2,433 |
| Length of inner labial setae | 15 | 13–20 |
| Length of outer labial setae | 35 | 30–35 |
| Length of cephalic setae | 19 | 12–14 |
| Mandible length | 18 | 13–17 |
| Mandible length to width, ratio | 2 | 1.7–2 |
| Tooth length | 8 | 7–10 |
| Distance from nerve ring from anterior end | 145 | 133−145 |
| Pharynx (oesophagus) length | 723 | 625–700 |
| Tail length | 183 | 167–198 |
| Anal body diameter | 41 | 39–43 |
| Spicule length as arc | 30 | 21–33 |
| Spicule length as arc / anal body diameter | 1 | 0.5–0.8 |
| Length of gubernaculum | 6 | 5–9 |
| a | 49.4 | 37.6–52.1 |
| b | 3.4 | 2.9–3.5 |
| c | 13.5 | 10–13.7 |
| c' | 4.5 | 4.1–5 |

**Table 4 Kimura 2-parameter distance between all available sequences of *Enoploides* species based on mtCOI alignment.**

| | | | | | | |
| --- | --- | --- | --- | --- | --- | --- |
| *Enoploides koreanus* [MN514234] | | | | | | |
| *Enoploides koreanus* [MN514235] | 0.000 | | | | | |
| *Enoploides koreanus* [MN514236] | 0.000 | 0.000 | | | | |
| *Enoploides koreanus* [MN514237] | 0.000 | 0.000 | 0.000 | | | |
| *Enoploides* sp. [HM564984] | 0.191 | 0.191 | 0.191 | 0.191 | | |
| *Enoploides* sp. [HM564985] | 0.236 | 0.236 | 0.236 | 0.236 | 0.218 | |
| *Enoploides* longispiculosus [FN998930] | 0.225 | 0.225 | 0.225 | 0.225 | 0.204 | 0.184 |

their accession numbers are MN514234–MN514242 (mtCOI) and MN512311–MN512318 (18S).

## DISCUSSION

One matter unresolved from this revision is the synonymy and confusion of species *Enoploides spiculohamatus*, *E. labiatus* and *E. longispiculosus*. Both *Wieser & Hopper (1967)* and *Benwell (1981)* suggested abandoning the issue as it cannot be proven. The main problem is that the original descriptions (*Bresslau & Schuurmans Stekhoven, 1940*; *Schuurmans Stekhoven Jr, 1935*) are poor and in the case of *Schuurmans Stekhoven Jr (1935)*, it may not even be *E. spiculohamatus* according to *Benwell (1981)*. As nothing can be done
**Table 5** Sequences retrieved from GenBank for phylogenetic analysis of this study.

| Name | Family | Marker | Accession number | Reference |
|------|--------|--------|------------------|-----------|
| *Metoncholaimus* sp. | Oncholaimidae | 18S rRNA | GU139748.1 | *Pereira et al. (2010)* |
| *Oncholaimus* sp. | Oncholaimidae | 18S rRNA | GU139747.1 | *Pereira et al. (2010)* |
| *Viscosia* sp. | Oncholaimidae | 18S rRNA | GU139749.1 | *Pereira et al. (2010)* |
| *Rhabdodemania* sp. | Rhabdodemaniidae | 18S rRNA | GU139750.1 | *Pereira et al. (2010)* |
| *Enoploides koreanus* | Thoracostomopsidae | 18S rRNA | MN512311 | This study |
| *Enoploides koreanus* | Thoracostomopsidae | 18S rRNA | MN512312 | This study |
| *Enoploides koreanus* | Thoracostomopsidae | 18S rRNA | MN512313 | This study |
| *Enoploides koreanus* | Thoracostomopsidae | 18S rRNA | MN512314 | This study |
| *Enoploides koreanus* | Thoracostomopsidae | mtCOI | MN514234 | This study |
| *Enoploides koreanus* | Thoracostomopsidae | mtCOI | MN514235 | This study |
| *Enoploides koreanus* | Thoracostomopsidae | mtCOI | MN514236 | This study |
| *Enoploides koreanus* | Thoracostomopsidae | mtCOI | MN514237 | This study |
| *Enoploides longispiculosus* | Thoracostomopsidae | mtCOI | FN998930 | *Derycke et al. (2010)* |
| *Enoploides* sp. | Thoracostomopsidae | 18S rRNA | GU139765.1 | *Pereira et al. (2010)* |
| *Enoploides* sp. | Thoracostomopsidae | mtCOI | HM564984 | *Bik et al. (2010)* |
| *Enoploides* sp. | Thoracostomopsidae | mtCOI | HM564945 | *Bik et al. (2010)* |
| *Epacanthion hirsutum* | Thoracostomopsidae | 18S rRNA | MN512317 | This study |
| *Epacanthion hirsutum* | Thoracostomopsidae | 18S rRNA | MN512318 | This study |
| *Epacanthion hirsutum* | Thoracostomopsidae | 18S rRNA | MG599065 | B Shi, S Pu & K Xu, 2017, unpublished data |
| *Epacanthion hirsutum* | Thoracostomopsidae | mtCOI | MN514241 | This study |
| *Epacanthion hirsutum* | Thoracostomopsidae | mtCOI | MN514242 | This study |
| *Epacanthion* sp. | Thoracostomopsidae | 18S rRNA | GU139759.1 | *Pereira et al. (2010)* |
| *Mesacanthion* sp. 1 | Thoracostomopsidae | 18S rRNA | GU139757.1 | *Pereira et al. (2010)* |
| *Mesacanthion* sp. 2 | Thoracostomopsidae | 18S rRNA | GU139756.1 | *Pereira et al. (2010)* |
| *Mesacanthion* sp. 3 | Thoracostomopsidae | 18S rRNA | GU139758.1 | *Pereira et al. (2010)* |
| *Bathylaimus* sp. | Tripyloididae | 18S rRNA | GU139751.1 | *Pereira et al. (2010)* |

about these species, they are left as valid species, but the fact that diagnostic features of *E. labiatus* and *E. longispiculosus* are nearly identical to one another remains problematic (Table 2).

The genus *Enoploides* currently consists of 27 valid species. The most recent review by *Smol, Muthumbi & Sharma (2014)* reported 28 valid species. It is unclear which exact species were listed as valid during their report, as a full species list was not provided. Aside from a list of doubtful species provided by *Wieser & Hopper (1967)*, one more species was transferred to species inquirenda through this revision. *Enoploides filicaudatum* is one of three species transferred from *Epacanthion* by *Greenslade & Nicholas (1991)*. While mandibles may resemble those seen in the genus *Enoploides*, the fact that the original description was based on two juveniles is problematic. *Mawson (1956)* pointed out that the distinguishing characteristic of this species is the shape of its tail. Such an ambiguous characteristic should not be used as sole reason to discern species, and thus it is regarded as species inquirenda.

## CONCLUSION

Our work reports *Enoploides koreanus* sp. nov., and *Epacanthion hirsutum* from Jeju Island, Korea. The new species belongs to a group with short spicules (<150 µm) and is closely related to *E. disparilis* and *E. cirrhatus* based on simplicity and similarity of the spicules and gubernaculum. However, these two species lack the pre-anal supplementary organ present in the new species. Their body ratios also differ to certain degree. Amplification success with the 18S rRNA gene allowed us to visualize the phylogenetic position of our species, which formed clades with their respective congeners with acceptable bootstraps. From compiling and comparing valid species of the genus, we also agree with *Wieser & Hopper (1967)*, that basing descriptions on females and juveniles is ill-advised for this genus, as both lack the distinguishable morphological characteristics evident in males, such as pilosity of the head, morphology of the spicule and gubernaculum, and presence or absence of the pre/post-anal supplementary organ. A bibliographic review of the genus has updated the diagnosis and list of valid species to 27, and a new key to the genus as well as tabular key comparing diagnostic characters of all valid species within the genus have been provided. *Epacanthion hirsutum* reported in Korea agrees with the original description provided by *Shi & Xu (2016)*. Its distinguishing character, a single row of densely arrange setae below the nerve ring, is easily discernible and there were no discrepancies with any of the measurements when compared with the original (Table 3).

**Abbreviations**

| | |
|---|---|
| **a** | body length/maximum body diameter |
| **abd** | anal body diameter |
| **b** | body length/pharynx length |
| **c** | body length/tail length |
| **calc** | calculated or measured from published measurements and/or figures |
| **cs** | cephalic setae |
| **c'** | tail length/anal body diameter |
| **ils** | inner labial setae |
| **ols** | outer labial setae |
| **scs** | subcephalic setae |

## ACKNOWLEDGEMENTS

We would like to thank Jaehyun Kim and Jisu Yeom for their insight on molecular analysis.

### Funding

This study was supported by a grant entitled "2018 Graduate Program of Undiscovered Taxa" from the National Institute of Biological Resources (NIBR) funded by the Ministry of Environment (MOE) of the Republic of Korea (NIBR201839201), grants (NRF-2017K2A9A1A06051528, NRF-2018R1D1A1B07050117) from the Korea Research Foundation, the BK21 Plus Program (Eco-Bio Fusion Research Team, 22A20130012352) funded by the Ministry of Education (MOE, South Korea), a grant (R2020035) from the National Institute of Fisheries Science (NIFS) of the Republic of Korea, and a grant 18-504-51026 Russian Fund of Basic Research. The funders had no role in study design, data collection and analysis, decision to publish, or preparation of the manuscript.

### Grant Disclosures

The following grant information was disclosed by the authors:
National Institute of Biological Resources (NIBR): NIBR201839201.
Korea Research Foundation: NRF-2017K2A9A1A06051528, NRF-2018R1D1A1B07050117.
Ministry of Education (MOE, South Korea): 22A20130012352.
National Institute of Fisheries Science (NIFS): R2020035.
Russian Fund of Basic Research: 18-504-51026.

### Competing Interests

The authors declare there are no competing interests.

### Author Contributions

- Raehyuk Jeong conceived and designed the experiments, performed the experiments, analyzed the data, prepared figures and/or tables, authored or reviewed drafts of the paper, and approved the final draft.
- Alexei V Tchesunov conceived and designed the experiments, analyzed the data, authored or reviewed drafts of the paper, and approved the final draft.
- Wonchoel Lee conceived and designed the experiments, authored or reviewed drafts of the paper, and approved the final draft.

### DNA Deposition

The following information was supplied regarding the deposition of DNA sequences:

Sequences of *Enoploides koreanus* sp. nov, unidentified *Enoploides* sp., and *Epacanthion hirsutum Shi & Xu, 2016* obtained in this study are available at GenBank: MN514234–MN514242 (mtCOI) and MN512311–MN512318 (18S).

### Data Availability

The raw measurements of *Enoploides koreanus* sp. nov. are available in Table S1. All specimens are deposited in National Institute of Biological Resources (South Korea). Holotype (NIBRIV0000858255), Allotype (NIBRIV0000858256) and seven Paratypes (NIBRIV0000858257-NIBRIV0000858263).

## New Species Registration

The following information was supplied regarding the registration of a newly described species:

Publication LSID: urn:lsid:zoobank.org:pub:6F60918D-9DE1-4B75-A251-C01E0694D01F

Enoploides koreanus sp. nov. LSID: urn:lsid:zoobank.org:act:87CC02F7-3A2E-4137-84E5-C8E7A97DA6D2.

## Supplemental Information

Supplemental information for this article can be found online at http://dx.doi.org/10.7717/peerj.9037#supplemental-information.

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
