# Peer review of "Two species of Thoracostomopsidae (Nematoda: Enoplida) from Jeju Island, South Korea"

_PeerJ, doi:10.7717/peerj.9037_

## Round 0.1 · original submission · Major Revisions

The manuscript has to be improved, particularly the Engish should be corrected to ensure that an international audience can clearly understand the text. Some new data are needed, for example to check some type or identified voucher specimens to confirm the taxonimical status of some species of Enoploides with debated validity. Other corrections have been suggested by reviewers.

Reviewer 1 ·

Basic reporting

Manuscript by Jeong, Tchesunov and Lee is devoted to description of one new species and one known species nematodes from the family Thoracostomopsidae. Important parts of the study are the revision of the genus Enoploides and identification key.
The English language should be improved to ensure that an international audience can clearly understand your text.
The list of reference is quite full, but there are some reference missed:
Ditlevsen 1919; Wieser & Hopper, 1957.
Introduction part is clear and explains well the aim of the study.
Drawings look like drafts and must be improved for publishing. The presence of SEM photo is a advantage for the paper. But light microscopical photo are highly recommended. It would not take a lot of time for authors as fas as they have made voucher photo for sequenced specimens.

Experimental design

Although it is very important to accumulate the data on DNA of free-living nematodes in GenBank, the size of 18S rRNA site amplified is only about 300 bp. Unfortunately, It is very short site for this gene. Such a short site is not enough to make any reliable conclusions on phylogeny of the group discussed. It is better to omit the part concerning phylogenetic tree and its discussion.
The descriptions are very detailed. Usually description of nematodes are made in telegraph style. For example, instead of "four shorter cephalic setae sharing one crown" (line 430) better use "six outer labial setae and four cephalic setae in one crown".
Line 438 "cervical somatic setae distributed along the body in singlets". Cervical setae can not be scattered along the body as fas as they are situated exclusively in cervical region. There is no need using term "in singlets" if the single setae scattered along the body. Better say "Cervical setae irregularly scattered along the cervical region"
Line 438 Amphid not observed. May be more careful examination of slides and SEM stubs will help?
Line 440 What does it means "pharynx annulated"?
Line 444 "82 mkm long above cloacal opening" Do you mean 82 mkm above cloacal opening?
Line 455 "Reproductive system didelhpic amphidelphic flexed inwards" the sentence is not clear. Please, clarify it in drawing as well. It does not look amphidelphic in drawing.
Key to valid species of Enoploides is very important and careful part of the manuscript. Line 548-549 What is post-anal cuticular element? There is no such elements in original drawings. May be the additional picture of this element would be helpful for manuscript readers.

The same comment to the secription of the second species, please, steak to telegraf style.
Line 607-608 "Base of mandibles widening near the base" Do you mean base of the base? Or just base?
Line 616. "Caudal glands protruding anterior to the anus" Better say "caudal glands in pre-anal position"

Validity of the findings

Conclusions are well stated, all underlying data have been provided.

Reviewer 2 ·

Basic reporting

.

Experimental design

.

Validity of the findings

.

Additional comments

See attachment

Annotated reviews are not available for download in order to protect the identity of reviewers who chose to remain anonymous.

Reviewer 3 ·

Basic reporting

Title: Two species of Thoracostomopsidae (Nematoda: Enoplida) from Jeju Island, South Korea

Reviewer #1
General Comments
In this study, the authors described a new species of Enoploides koreanus sp. nov. and redescribed Epacanthion hirsutum Shi & Xu 2016, both belonging to the family Thoracostomopsidae (Nematoda: Enoplida). Some genetic data of the two species were also reported. The genus Enoploides was reviewed based on references and a key to species was provided. The illustrations and SEM micrographs are of good quality. However, there are a number of serious flaws: (i) The name of the new species is not correct, because the generic name Enoploides is a feminine gender, but specific name “-koreanus” is masculine gender, which are conflicted. (ii) The 18S rDNA with very slow evolutionary rate is not suitable for species identification of nematodes. So Table 7 has no any scientific significance, that could not support your specimen collected from Jeju Island to be congeneric with Epacanthion hirsutum. (iii) The method for phylogenetic analysis is very rough (lacking of details). Which group was chosen as outgroup? Most species of Enoploides with 18S sequence data available in GenBank, i.e. Enoploides delamarei, Enoploides longispiculosus, Enoploides sp. 161216.19 etc. have not been included in the phylogeny. (iv) the authors did not examine any type or identified voucher specimens of Enoploides species for revision of this genus, thus the reliability of results is weak. I strongly suggest the authors check some type or identified voucher specimens to confirm the taxonimical status of some species of Enoploides with debated validity. (v) the English of manuscript needs much improvement. The text (i.e. abstract, Introduction, description) is wordy and has some grammatical errors. It would benefit from a revision made by a native speaker.

Specific comments
1. Abstract: the present Introduction is rather wordy. Some sentences have obvious grammatical errors. Please rewrite it.
2. Please do not use the abbreviation in abstract, give the meaning of the letter ”b”.
3. Introduction, Line 81-82: The aim of this study was to review and revise the genus Enoploides while reporting a new species....you did not revise the genus Enoploides?
4. Materials and Methods: Which group was chosen as outgroup for the phylogeny? How many bootstrap replications did you test?

Experimental design

(i) The authors used the 18S rDNA as genetic marker for speciese identification in the present study. However, the 18S rDNA with very slow evolutionary rate is not suitable for species identification of nematodes. So the result of Table 7 could not support the specimen collected from Jeju Island to be congeneric with Epacanthion hirsutum. (ii) The method for phylogenetic analysis is very rough (lacking of details). Which group was chosen as outgroup? Most species of Enoploides with 18S sequence data available in GenBank, i.e. Enoploides delamarei, Enoploides longispiculosus, Enoploides sp. 161216.19 etc. have not been included in the phylogeny.

Validity of the findings

(i) the authors did not examine any type or identified voucher specimens of Enoploides species for revision of this genus, thus the reliability of results is weak. I strongly suggest the authors check some type or identified voucher specimens to confirm the taxonimical status of some species of Enoploides with debated validity.

Reviewer 4 ·

Basic reporting

I found the paper very orginal, well written and structured. I think that this type of taxonomical papers is really necessary because it is complete and increases our knowledge on the diversity of an important nematode family. In particular, I have appreciated that the authors analysed their samples with the aid of several tecniques (i.e. SEM, light microscopy and molecular approach) giving a complete description of the new species. Furthermore, I think that the new indentification key, theupdate list of valid species and new molecular insights will be important for the taxonomists of this phylum. Thus, I suggest the pubblication of this paper after minor revisions.
I reported in the file some minor changes that I suggest to the authors. The main ones are a doubt in the identification key and the suggestion to add further details in the SEM picture of the male of Enoploides koreanus

Experimental design

The approaches used are fully in agreement with the current taxonomical standards.

Validity of the findings

It is one of the most complete taxonomical paper that I have reviewed and I think that it will have a high relevance in the field.

Additional comments

lines 307-308; 312-313; 337-338; 342-343; 363-364; 368-369; 373-374; 390-391. I suggest to revise the english of this sentences;
line 411 Please add the total number of specimens analysed
line 530 Please check this point also considering the point 3 (see line 527) in which you stated that pre anal supplements were present ;
line 643: Change as follows "primer sets utilized in the present study."
line 673 delete italic in the following word "Mawson"
Fig 1 is it possible to add a drawing with the spicule and gubernaculum details?
Fig 3 B in my opinion it is quite difficult to see setae. Could you add a detail increasing the magnification?
Fig 3 C Also in this case I suggest a detail of the tip

Annotated reviews are not available for download in order to protect the identity of reviewers who chose to remain anonymous.

---

## Round 0.2 · Major Revisions

Dear Authors,

The article is now close to Acceptance. However, the Section Editor has commented on your paper: As it is a taxonomic species describing new species, there are terms are formats that have special meaning that are not used correctly in it. For example, the first authority in the Abstract is missing a comma - a small but important detail. Also, the term "new species" should not be really used for past papers - "they described" etc is safer, reserving "new" for new species in THIS paper. As well, you cannot "describe" in taxonomy a species already named, as you now state in the first part of the Abstract. In short, this paper should be checked by an ICZN-familiar taxonomist.

May I suggest that you seek a colleague who knows ICZN issues to look at it for you and correct the errors.

Reviewer 1 ·

Basic reporting

I am satisfied with revision of manuscript. It could be accepted for publication.

Experimental design

Methods described with sufficient detail & information to replicate.

Validity of the findings

All underlying data have been provided; they are robust, statistically sound, & controlled.

Additional comments

Authors said that they added Phillips et al., 2016 to line 106, but I have not found it. It is also absent in reference list.

Reviewer 3 ·

Basic reporting

The language of Ms is clear now. The introduction is good enough.

Experimental design

The research question has been well solved. The methods used in the study is suitable and powerful.

Validity of the findings

The authors described a new species of Enoploides koreanus sp. nov. and redescribed Epacanthion hirsutum Shi & Xu 2016, both belonging to the family Thoracostomopsidae (Nematoda: Enoplida) and provided important morphological and genetic data for this group.

Additional comments

This revised version of manuscript has been significantly improved. All of my recommendations have been complied. I have no additional comments.

---

## Round 0.3 · Minor Revisions

Your manuscript still needs some modifications which are marked in a PDF from Reviewer 2.

Reviewer 2 ·

Basic reporting

See comments in the annotated PDF.

Experimental design

See comments in the annotated PDF.

Validity of the findings

See comments in the annotated PDF.

Additional comments

See comments in the annotated PDF.

Annotated reviews are not available for download in order to protect the identity of reviewers who chose to remain anonymous.

---

## Round 0.4 · accepted · Accept

After reviewing the rebuttal letter and the new version of your manuscript I'm pleased to confirm that your paper has been accepted for publication in PeerJ.

Thank you for submitting your work to this journal.